# Overview of Existing Heat-Health Warning Systems in Europe

**DOI:** 10.3390/ijerph16152657

**Published:** 2019-07-25

**Authors:** Ana Casanueva, Annkatrin Burgstall, Sven Kotlarski, Alessandro Messeri, Marco Morabito, Andreas D. Flouris, Lars Nybo, Christoph Spirig, Cornelia Schwierz

**Affiliations:** 1Federal Office of Meteorology and Climatology MeteoSwiss, 8058 Zurich Airport, Switzerland; 2Meteorology Group, Dept. Applied Mathematics and Computer Sciences, University of Cantabria, 39005 Santander, Spain; 3Centre of Bioclimatology (CIBIC), University of Florence, 50144 Florence, Italy; 4Department of Agriculture, Food, Environment and Forestry (DAGRI), University of Florence, 50144 Florence, Italy; 5Institute of BioEconomy—National Research Council, 50019 Florence, Italy; 6FAME Laboratory, Department of Exercise Science, University of Thessaly, 42100 Trikala, Greece; 7Department of Nutrition and Exercise Sciences, University of Copenhagen (NEXS), 2100 Copenhagen, Denmark

**Keywords:** heat-health warning system, action plan, intervention strategy, user-tailored, heat stress

## Abstract

The frequency of extreme heat events, such as the summer of 2003 in Europe, and their corresponding consequences for human beings are expected to increase under a warmer climate. The joint collaboration of institutional agencies and multidisciplinary approaches is essential for a successful development of heat-health warning systems and action plans which can reduce the impacts of extreme heat on the population. The present work constitutes a state-of-the-art review of 16 European heat-health warning systems and heat-health action plans, based on the existing literature, web search (over the National Meteorological Services websites) and questionnaires. The aim of this study is to pave the way for future heat-health warning systems, such as the one currently under development in the framework of the Horizon 2020 HEAT-SHIELD project. Some aspects are highlighted among the variety of examined European warning systems. The meteorological variables that trigger the warnings should present a clear link with the impact under consideration and should be chosen depending on the purpose and target of the warnings. Setting long-term planning actions as well as pre-alert levels might prevent and reduce damages due to heat. Finally, education and communication are key elements of the success of a warning system.

## 1. Introduction

Climate change and, in particular, the increase of mean global temperature has become a major concern for society. A direct consequence of higher mean temperatures is the increase of the frequency of natural hazards [1] and, in particular, the increase of the frequency and duration of heat waves [2,3,4]. Those changes, depicted by past observations and future projections, constitute a robust signal of climate change [5]. Moreover, changes in temperature extremes (with stronger impacts on society) are projected to be larger than those in mean temperature [6,7]. Summers as hot as 2003 in Europe (statistically extremely unlikely) could happen every other year by the end of the 21st century in Central Europe [2], and even earlier in Southern Europe, with corresponding consequences for human heat exposure [4]. In light of this situation, it is crucial to develop strategies to mitigate the societal effects of changes in extreme heat conditions at different temporal scales, from the next days to decades, including monitoring and warning systems.

Different factors need to be taken into account in order to assess the impacts of changing frequencies and intensities of heat waves. From the meteorological point of view, the consideration of all relevant meteorological variables is essential. Although there is no unique definition of heat waves, the World Meteorological Organization recommends to consider more than five consecutive days with a maximum temperature anomaly exceeding 5 °C [8]. Other factors, such as relative humidity and solar radiation, are often overlooked. The combination of high temperature and humidity invokes heat stress and, precisely, humid heat is directly related to human thermoregulation (Ref. [9] and references therein) and, therefore, to human (dis)comfort. The effect of solar radiation increases heat exposure considerably, which might lead to critical heat stress conditions in Southern and Central Europe by the end of the 21st century [10]. The projected changes in heat stress conditions [11,12,13] relate to increased mortality [14] and labor productivity losses [10,15,16,17,18]. Even under a 2 °C warmer world, the number of heat-related deaths in urban areas could multiply by a factor of 2 to 3 compared to current climate [19]. Considering health-related metrics (combining temperature and humidity, for instance) provides also more robust climate projections, since uncertainties in the combined index are smaller than if uncertainties in the two variables were independent [10,20]. Another relevant meteorological variable which is often neglected in the heat wave definition is minimum temperature. Under high minimum temperatures people might not recover from the daytime heat and might subsequently not be able to handle any extreme heat the following day [21]. This is especially relevant in cities, which often cannot cool down at night due to their specific radiative, thermal, moisture and aerodynamic properties [22] leading to the so-called urban heat island (UHI) effect. While the response to a warmer climate in urban and rural locations is similar, the occurrence of nights with extremely high heat stress increases more in cities than surrounding rural areas [23].

As a response to the projected intensification of heat conditions but also as an important component of strategies to cope with present-day climate variability including extremes, the improvement and development of heat-health warning systems (HHWSs) is among the priorities of the World Meteorological Organization (WMO) and the World Health Organization (WHO). An HHWS is a system that combines meteorological forecasts and public health actions in order to reduce heat-related impacts on human health during hot conditions [24,25]. HHWSs build upon the establishment of certain thresholds of human-health tolerance to extreme weather. According to Ref. [26], a HHWS requires: (1) reliable meteorological forecasts for the population or region of interest, (2) robust understanding of the cause-and-effect relationships between the thermal environment and health impacts, (3) effective response measures to implement within the window of lead-time provided by the warning, (4) the involvement of institutions and civil society, with sufficient resources, capacity, knowledge, and political will to undertake the specific response measures. The success of such HHWSs relies on addressing awareness, preparedness, organizational issues, and actor networks in a proactive and focused manner [19].

Previous works gathered the characteristics of existing heat warning systems in Europe, highlighting their heterogeneity [24,27]. Here we present a state-of-the-art review of HHWSs and heat-health action plans (HHAPs) of 16 countries, based on existing literature, web research and direct contact to National Meteorological and Hydrological Services (NMHSs). A few examples outside of Europe are also presented for comparison purposes. Among the examined characteristics of the HHWSs, we identify the forecast system (including details on the variables and threshold for triggering the warnings), the various components of action plans (including intervention strategies and evaluation/revision of the warnings) and the communication of the warnings (including notification strategies to users and stakeholders). This work paves the way for future HHWSs, such as the one currently under development in the framework of the Horizon 2020 HEAT-SHIELD project, specifically addressed to occupational sectors (www.heat-shield.eu, Refs. [28,29]).

The present paper covers the historic and scientific background for developing HHWSs and HHAPs (Section 3), reviews the existing HHWSs and HHAPs in Europe with comparisons across countries and reference to other global systems (Section 4), and provides recommendations for the development of future or the improvement of current heat-health alert and advice systems (Section 5).

## 2. Methodology

A review of existing literature on warning systems and a web-search of the websites of NMHSs were conducted between January and May 2019. The Web of Science database was searched using the terms “heat”, “warning system” and “health”, resulting in the retrieval of 176 papers. The web search consisted of a screening of NMHSs web sites and gathering of WMO and WHO reports by two independent reviewers (A.C. and A.B.). Only documents and publications in English, Spanish and German were considered. Additionally, questionnaires were sent personally to contacts at NMHSs in order to add expert knowledge. These questionnaires consisted of four parts: the weather forecast system (Section 4.1 Weather Forecast Systems), the heat-health warning system itself (Section 4.2 Heat-Health Warning Systems), action plans and intervention strategies (Section 4.3 Intervention Strategies) and the communication and dissemination of the warnings (Section 4.4 Communication and Dissemination of the Heat Warnings).

In total, we gathered information from 16 European countries (see Table 1).

### 2.1. Weather Forecast System

Different European Meteorological Agencies use different meteorological forecast systems, they either run their own simulations or use those from the European Center for Medium-Range Weather Forecasts (ECMWF) or a combination of both. Going from short- to medium- and long-range forecast implies larger uncertainties and often requires the need for an ensemble of simulations (based on multiple realizations representing equally realistic situations). Such considerations were addressed in the questionnaire (see Table 1):For which country or region is the forecast produced?Which is the provider institution?What is the model or forecast system producing the forecast? Is it deterministic or probabilistic (i.e., ensemble forecast)?What is the spatial resolution of the warnings, i.e., city/regional/country level?What is the temporal resolution of the forecast, i.e., how often are the updates made?What is the lead time, i.e., how much time in advance are the warnings issued?

### 2.2. Heat-Health Warning System

Once the forecast system is set, a warning system needs to be defined in order to protect the population against meteorological hazards. Those warnings are usually associated with different risk levels and are triggered when a specific variable, index or a combination of variables exceeds certain thresholds. In this work, we focus on heat warnings, which are triggered based on a variety of variables and thresholds. Likewise, there is no uniform definition of a heat wave [8,21]. The following questions were used to assess this aspect:Which index or variable is used for the heat warnings?Which threshold(s) triggers the warnings or warning levels?What is the rationale behind the threshold used to issue heat warnings? Is it based on specific epidemiological studies, climatological percentiles or taken from literature?In addition to the general public, are there other specific target groups?

### 2.3. Heat-Health Action Plans

Nowadays, heat warnings are not an isolated part of the National Meteorological Services. Instead, they are being included as part of heat-health action plans, in collaboration with other institutions. The interdisciplinary approaches are a key element within meteorological and climate services. To assess the characteristics of the HHWSs and HHAPs, we examined the following questions (Table 3):Are the heat warnings part of a heat action plan?If yes, which are the intervention strategies (i.e., actions taken based on the warning level)?Is the warning system subject to evaluation and revision? How and how often?

### 2.4. Communication and Dissemination of the Heat Warnings

The above-mentioned forecasts, warnings and actions as a response to heat would not be useful without organized communication and dissemination strategies. Those refer to the communication to the general public, to specific target groups, as well as to communication among different institutions or governmental bodies. In this sense, the following questions were considered (Table 4):In which language are the warnings delivered?Which kind of notification system is used to inform target groups? (i.e., do they receive any user-oriented notification and, if yes, by which means?)How is the information provided to stakeholders?

## 3. Existing Warning Systems and Action Plans

### 3.1. Evolution of Warning Systems

Heat-health warning systems (HHWSs) are the main response to heat waves worldwide and their aim is to alert decision makers and the general public of dangerous heat situations [19]. They consist typically of weather forecasts, methods to assess the weather-health relation, a system of graded alerts and the communication of such alerts. Regarding weather forecasts, important developments have been continuously made, increasing the skill of the numerical models thanks to the better representation of physical processes, improvements in the data assimilation system, the use of denser and better-quality observations, etc. For instance, for 500 hPa geopotential height (a variable that describes the large scales in the free atmosphere) forecasts have improved by about 1.5 days per decade [30]. Today’s 5-day forecasts from the European Center for Medium-Range Forecasts (ECMWF), for example, are as skillful as 3-day forecasts were in 2001.

The variables on which HHWSs in Europe are based on and the thresholds triggering the warnings are very diverse. The most straightforward way is to consider the exceedance of a threshold of daily maximum temperature on a given day. The thresholds for maximum temperature are region-specific, ranging from 30 °C in Belarus to 38 °C in Greece [31], in Europe, and up to 45 °C in Phoenix (USA). Some HHWSs include a minimum duration of the heat event (e.g., France, Germany) and/or are based on a combination of several meteorological variables or indices (e.g., the heat index in Switzerland).

Regarding the methods to assess the weather-health relation, the first component is to choose the variables which describe that relation and define the threshold(s) of weather variable(s) linked to the impact (e.g., in epidemiological studies) or based on climatological extreme values (e.g., high percentiles). The thresholds trigger the warnings, i.e., they establish a system of graded alerts, which is often related to specific actions. The communication of the warnings refers, on the one hand, to the general population and specific target groups about the severity of the event and, on the other hand, to advise governmental agencies about the severity of health impacts [19]. Heatwave levels are either graded according to the intensity of the heatwave, the duration or, in some cases, a combination of both. This way several alert levels is defined differently in every country. Differences are also found in the pre-alert levels, which are sometimes active during the whole warning season and sometimes several days before an event.

HHWSs were mostly developed as a response to extreme events (the first one in the city of Philadelphia, PA, USA, in 1995). In some North American cities, synoptic-based heat watch–warning systems were developed between 1995–2005 [32]. Only one HHWS was operational before 2001 in Europe, namely in Lisbon. In 2001, the WMO chose Rome as a pilot city for the development and implementation of an air-mass-based HHWS (which identifies oppressive air masses associated with an increase in mortality). The 2003 heat wave had terrible consequences in Europe, in terms of mortality (more than 70,000 excess deaths across 12 European countries, Ref. [33]), forest fires, impacts on water resources, power cuts, transport restrictions and agricultural losses, which amounted to more than 13 billion euros (de Bono et al. 2004). As a consequence, many other European countries started to implement their HHWSs after summer 2003 [25,27,34]. In Switzerland, the HHWS was created in 2004 as a response to requests by the Cantons Ticino and Geneva. Due to the lack of an epidemiological study in this country, an internationally used heat stress index, the heat index [35,36], was adopted to set the basis of the HHWS. The Netherlands developed a HHWS and action plan in 2007, at the request of the Ministry of Health, Welfare and Sport, and has been used every year since then. The number of European countries with implemented HHWS has continuously increased, even in northern Europe. By 2006, HHWS were operational in 16 countries and by 2009 in 28 countries [37]. In Sweden, the NMHS introduced heat warnings in 2013 and the thresholds triggering the warnings were largely based on a study by the Department of Public Health and Clinical Medicine of the University of Umeå, which was focused on the relation between the temperature and the rate of mortality. At a continental scale, MeteoAlarm (www.meteoalarm.eu) represents the joint official website relying on information from Europe’s national weather services. It was developed by EUMETNET (the network of public European weather services who are members of the WMO) and provides advice on exceptional weather, including extreme heat.

### 3.2. Heat-Health Action Plans

A HHWS is the weather-based alert component of a wider heat-health action plan (HHAP). According to Ref. [38], other components of a HHAP are the identification of vulnerable population groups, interaction with stakeholders, design and operationalization of intervention strategies, implementation of longer-term heat-mitigation procedures (e.g., public education and urban planning and design) and evaluation of HHWS effectiveness. The strategy elements of the HHAP are very diverse and comprise: education and raising awareness; preparedness and guidance in order to avoid heat exposure; communication plans; evaluation programs; health surveillance system; and advice on longer term strategies for reducing heat risk [25]. HHAPs usually join together the national weather services and health authorities. Other important agents in different stages of the HHAP are general practitioners and health centers, civil protection and social services [25].

Despite the implementation of HHWSs and HHAPs in European countries after the 2003 heat wave, the consequences of other heat episodes (e.g., deaths in summer 2006) confirmed that the efforts were not enough and that there was a need for further developments and revisions of the systems [39]. On the other hand, the resilience and acclimatization capacity of the population increases with climate change, therefore the HHWSs should account for such changing vulnerability patterns and adaptation [38]. Otherwise, warnings based on current thresholds will be issued more often in a warmer world, hence, they might lose credibility.

Since 2003 also research organizations and research projects have dedicated efforts to the analysis and development of HHWSs. In Tuscany (Italy), a study conducted in 2012 highlighted a decrease of the heat effect on mortality after 2003 probably due to the implementation of a regional HHWS with specific interventions and preventive measures for safeguarding the health of the “frail elderly” [40]. The EuroHEAT project (2005–2007), coordinated by the WMO Europe and funded by the European Commission, quantified the health effects of heat in European cities and identified options for improving health systems’ preparedness for and response to the effects of heat waves. In the framework of this project, an interactive tool showing the probability of experiencing a heat wave in Europe was developed. It is based on probabilistic forecasts of the ECMWF up to ten days ahead (www.euroheat-project.org/dwd/). While this is a good example of a coordinated action to increase awareness, no actions or recommendations complete those meteorological warnings. Within the European Project HEAT-SHIELD (www.heat-shield.eu) a web platform has been developed to support protection of the European workforce in the main strategic European industry sectors (manufacturing, construction, transportation, agriculture and tourism) against heat. The heat warning system allows stakeholders to develop timely and precise prevention strategies and better planning of the work activities up to four weeks in advance, based on the medium and extended range ensemble forecast from the ECMWF (http://heatshield.zonalab.it). Warnings are issued based on user-tailored thresholds and some recommendations regarding amount of the drinking water and duration of work-rest cycles are provided along with the warning (see Ref. [29] in the present Special Issue for details).

## 4. Results

### 4.1. Weather Forecast Systems

The considered European countries use different forecast model systems (Table 1). In most of the countries, a high-resolution model (~2 km) runs for short lead times and the ECMWF probabilistic ensemble is used for longer lead times, or as a complementary tool to support the forecasters. There are ready-to-use products from the ECMWF such as the EFI (extreme forecast index) of temperature, indicating how extreme predicted temperatures are with respect to the forecast model climate. This is, for instance, used as a supporting tool in Switzerland. In all the analyzed countries, warnings are provided at regional level (regions, cantons, counties, provinces) and even for cities in the case of Lisbon and several Italian cities. They are updated daily or several times a day and they are issued with lead times ranging from two to eight days.

### 4.2. Heat-Health Warning Systems

The analyzed HHWSs are based on either single- or few-parameter methods (i.e., they use a single metric of temperature or a combination of several variables in one index). In most countries (Table 2) warnings are based on maximum or mean temperature only (Greece, Hungary, Netherlands, North Macedonia, Portugal, Romania, Slovenia, Sweden). Some others consider maximum and minimum temperature values to trigger the heat warnings (England, Belgium, France and Spain) and a few include other variables such as humidity (Switzerland, Italy, Austria and Germany). Moreover, warnings in Germany and Austria are based on the combination of a thermal-stress index (perceived temperature, based on temperature, radiation, wind, humidity) with minimum temperature. It is also interesting to note that Portugal and Spain present a heat wave definition based on surpassing certain maximum temperature thresholds, but the alert levels rely on more sophisticated systems (ÍCARO index—Importância do Calor e a sua Repercussão nos Óbitos [importance of heat and its repercussion on deaths]—in Portugal and a combination of maximum and minimum temperature in Spain). In most of the countries, the variable that triggers the warnings is exclusively used for hot conditions, whereas perceived temperature in Germany is used also to quantify cold stress [41].

In Greece, the warnings issued by the Hellenic National Meteorological Service rely on maximum temperatures (and the heat index as a supplementary tool), whereas the National Observatory of Athens operates a system for forecasting human-biometeorological conditions (www.meteo.gr), based on the Physiological Equivalent Temperature (PET, Ref. [42]). This physiological model is optimized for a standard European 35-year-old male, 1.75 m tall, weighing 75 kg, and having a metabolic heat production rate of 80 W (light activity) and includes an adaptive clothing model, based on air temperature, which is used for parameterizing clothing insulation. The Italian system is the only one in our collection based on an air mass model, thus considering further the atmospheric conditions related to extreme heat. It is also based on the maximum apparent temperature approach (Tappmax) and presents a variety of regional approaches.

In most countries, the reference values or thresholds are determined from epidemiological studies which link heat to mortality data, i.e., by modeling the temperature-mortality relationship. For this purpose, several statistical packages are available, such as the widely used distributed lag nonlinear model [43], which has been used in several studies (e.g., Refs. [44,45,46]). In most cases, total, all-cause mortality data are used, whereas non-accidental mortality and mortality of those aged 65 and older (more vulnerable) are considered in Italy [47] and Toronto [48], respectively. In Lisbon, the ÍCARO system uses an algorithm derived from the relationship between maximum temperature and mortality; a 31% rise in mortality sets the threshold for an announcement level and a 93% for an alert level (Ref. [49], see Table 2). Concerning the synoptic-based system (air mass approach), the standardized mean summer mortality associated with each air mass is determined and air masses with statistically significant mortality greater than normal are identified [38]. In particular, air mass algorithms have been developed to account for differences in the air mass character from day to day, as well as their seasonality and persistence. The HHWS in Germany and Austria is, however, based on heat-budget models which evaluate the thermal stress on a typical human and bases thresholds on the different levels of stress. Although it is not explicitly based on a mortality response, a clear correlation with mortality has been established [50]. The thermophysiological assessment is made for a standardized person (a 35-year-old man, is 175 cm tall and weighs 75 kg), referred to as *Klima-Michel*, who adapts his clothing between 0.5 and 1.75 clo (clothing insulation unit). The assessment procedure is designed for staying outdoors [51].

In most of the systems, the thresholds are defined at the country level (i.e., same reference value for a specific warning level for the whole country). However, in England, France, Germany, Greece, Italy, North Macedonia and Spain they depend on the region. In Switzerland, the thresholds are the same for the whole country, but there are large differences in the state of HHWSs and HHAPs depending on the canton. Cantons Ticino and Geneva, with higher climatological heat exposure, have developed more detailed action plans (Table 3). Warning thresholds might present certain seasonality, being different at the beginning and at the end of the summer. For instance, in Spain the thresholds vary spatially and temporarily throughout the course of the year. Also, the German system accounts for short-term acclimatization within its calculation of the threshold of perceived temperature, since it considers the conditions in the previous 30 days [50].

The alert levels of a warning system are also set very differently across Europe (Table 2). In most countries two to four levels are used, denoted either numerically or with a colored scale (e.g., green, yellow, orange, red). Interestingly, in Sweden there are additionally “notification” and “risk” levels. The notification level works as a pre-alert level and the risk level relies on the same thresholds as the highest warning level but with higher model uncertainty. This highlights the need for working with probabilistic forecasts and shows a way of communicating this additional dimension.

Since all the considered HHWSs have been developed by the national weather services in European countries, the target groups are very wide (Table 2). The general population and specific vulnerable groups (elderly, homeless, children) are the main focus. Workers represent an important part of the population potentially at high-risk of heat exposure for many easily understandable reasons, with potential consequences for their health and work productivity. Despite this, few HHWSs target this group in Europe [52]. To reach them, it is necessary to have a good communication with other administrations (e.g., national health service), civil protection, NGOs, etc., but also directly with nursing facilities, hospitals and schools. Germany, the Netherlands and Switzerland explicitly mention workers and trade unions among the target groups.

### 4.3. Intervention Strategies

The purpose of the HHAPs is to define concrete and useful intervention strategies. The timing of the specific interventions to implement should be tied to the levels of warning used in the HHWS, with educational messages provided early enough [38]. The interventions can be classified into individual- and community-level responses (including responses by employers). Regarding the individual level, some of the common advice is to stay out of the heat, drink water, cool yourself down and keep the environment cool and look out for others [53]. In terms of keeping the environment cool, it is relevant to communicate that when it is very hot and dry, and when body core temperatures exceed 38 °C, using a fan alone actually increases heat stress, because of the limits of conduction and convection [38]. Regarding community-level actions, the main strategies are: develop educational campaigns, set-up dedicated telephone services, increase media announcements, contact the health and fire departments and social services to assist people with limited resources, use a registry of vulnerable people, who are visited at home and evacuated, if necessary, and distribute fans [25].

Some intervention strategies are defined by national and international organizations such as the International Labour Organization (ILO), which, in particular, establishes threshold limit values for occupational heat exposure [54,55]. Some of the recommendations specific to working people are: (1) Changing work practices, such as providing plenty of drinking water, scheduling heavy work during the cooler parts of the day, or reducing the physical demands during the hottest part of the day; (2) Alternate work and rest periods, with rest periods in a cool area; (3) Wearing appropriate clothing. Some more sophisticated engineering solutions being investigated nowadays include personal protection through movable personal microclimate cooling and technical developments in clothing based on cooling with ventilation (e.g., ventilated clothing with integrated electric fans) and phase change materials that absorb or release latent heat when they change phases [56]. There is a trade-off between the sustainability, appropriateness and feasibility of the different strategies and solutions against heat, and they need to be evaluated for each specific sector.

In most of the considered European countries, there is a HHAP which coordinates the actions after a heat warning is issued (Table 3). In terms of intervention strategies, preparation, monitoring and raising awareness through educational campaigns is crucial during the lower alert levels (or even during the whole summer season). Once there is a heat wave, proper communication with the health services and civil protection is as important as the provision of bottled water and cooling and sheltered areas. All countries include the preparation and adaptation of hospitals and staff in their active measures. A phone hotline is also a common strategy in most of the European countries. The reader is referred to Table 3 for a summary of the intervention strategies.

It is difficult to evaluate the success of HHWSs and HHAPs, due to the rarity of the events [25]. Regarding the accuracy of the meteorological approach, it is advisable to measure the predictive value of the meteorological threshold. There is a variety of statistical techniques to test the robustness of the weather-health predictive model, as well as the meteorological forecasts themselves. Other aspects prone to evaluation of the HHAPs are simplicity in the structure and operation system, acceptance by the participating agencies and stakeholders, sensitivity in terms of hit and false alarm rate), timeliness (timely warnings and responses), effectiveness of the response measures [57] and specificity of the forecasts to avoid diminishing the credibility of the forecasts [25].

The revision of HHWSs and HHAPs might include the adjustment of the thresholds triggering the warnings, for instance, due to the acclimatization ability under a warmer climate. However, that is not the only factor boosting thresholds updates, the prominence of air conditioning changes over time too. Ref. [58] suggests that better infrastructure development and greater access to air conditioning in Hong Kong might be associated with a higher, mortality-related threshold than in other Asian cities.

The revision might include, in the long-term, the effect of changes in city architecture. Building passive systems makes it possible to effectively heat or cool buildings by means of renewable energy [59]. Other changes in urban planning, such as more green spaces within the urban core, could help to alleviate the UHI effect [60]. For instance, urban green spaces are also able to influence the surrounding area by the urban green space cooling effect [61].

In most of the considered European countries, the HHAPs are revised annually (Table 3). In Sweden and Switzerland, they are evaluated after every warning and in Portugal only after extreme years.

### 4.4. Communication and Dissemination of the Heat Warnings

According to Ref. [38], the risk associated with extreme heat has to be communicated precisely and adjusted for the target group. Warnings and forecasts need to be not only understandable but also attractive in order to evoke interest and motivation to consume the information and take appropriate action. The WMO also identifies the importance of effective communication skills of the staff of the National Meteorological Services. Likewise, skills related to the interaction with the media, such as writing effective press releases and holding interviews, press conferences and press briefings, are considered highly important.

The review in Ref. [24] highlights that in most European countries, the HHWSs did not include any intervention apart from issuing a passive warning to the general public and to the local public health agencies, and a few issued warnings solely through the mass media or only to health agencies. This situation has changed in the recent years. Nowadays, HHWSs use a range of means and channels for the communication of heat warnings. Most of the European HHWSs provide plenty of information through their websites, mobile apps and social media (e.g., Twitter). Also, brochures, flyers and newsletters are sent to hospitals, nursing facilities and general practitioners as common practice. In addition to this, the Dutch system has a heat toolkit with information and communication tools about the risks of hot weather conditions. It contains a list of questions and answers, a public brochure, sample letters, a press announcement and the text of the National Heat Plan. In Spain, in addition to the usual communication channels, bulletins and Common Alerting Protocol (CAP) are also available via internet. The latter allows the introduction of probabilistic information, the description of the potential impacts and recommendations for the population with a standardize format (in xml format) and wording [62]. The data from the Spanish system is also available from an OpenData server for further analysis.

Regarding communication (Table 4), it is characteristic that all information and warnings are issued in the local language of each country and only Sweden, Hungary and Switzerland provide notifications in both the local language and in English. This is an important limitation, especially for South European countries, which receive millions of tourists every summer.

### 4.5. Comparison with Non-European National Heat-Health Systems

A few examples outside of Europe are also considered for the sake of comparison. The United States National Oceanic and Atmospheric Administration (NOAA) issues heat warnings for the entire United States based on the mean heat index (same as, e.g., Switzerland). The mean heat index is calculated as the average of the values from the hottest and coldest times of the day. Therefore, it is more representative of the entire 24-h period than a single daily maximum value. Forecasts are provided routinely for conditions 3 to 7 days in advance on the web site of the NOAA. When high daily mean heat index (105–110 °F, i.e., 40.6–43.3 °C, depending on local climate) is forecasted for at least 2 days and night time air temperatures above 75 °F, approximately 24 °C, (https://www.weather.gov/safety/heat-index), a warning is issued to the public and relevant agencies [24]. Those conditions are rare across much of the northern part of the United States and heat-related mortality occurs well below such thresholds. For this reason, several local offices have modified the thresholds and consider air mass models in order to account for local climatological conditions [32]. Those localized HHWSs have been initiated and partly financially supported by utility companies [32,38].

In addition to heat warnings, other risk levels are defined in the United States. Heat watches are issued when the risk of a heat wave has increased but its occurrence and timing is still uncertain; heat advisories are issued within 12 h of the onset of extremely dangerous heat conditions (heat index above 100 °F, approximately 38 °C, for at least 2 days and night time air temperatures above 75 °F), and heat outlooks are issued when the potential exists for an excessive heat event in the next 3–7 days (www.weather.gov/safety/heat-ww).

The Australian Bureau of Meteorology (www.bom.gov.au/info/thermal_stress) makes use of the apparent temperature (combination of temperature, humidity, wind speed and radiation absorbed by the human body) and the simplified wet bulb globe temperature (sWBGT, combination of temperature and humidity). One of the stakeholders of the Australian heat warnings is “Sports Medicine Australia”, who recommends limiting the intensity and duration of the sport events if sWBGT is between 26–29 °C, and postponing them if sWBGT is above 30 °C (see Ref. [63]). In Melbourne, a simple warning system based solely on daily mean temperature has been developed [64]. Daily mean temperature over 30 °C is related to an increase of 15–17% of the mortality of elderly people (65 years or more). Interestingly, daily maximum temperature does not have an important effect on the number of fatalities. Heat wave planning was identified as a priority by the government of the state of Victoria, where a detailed HHAP was developed in 2006 [65]. 13 pilot programs were funded for the implementation of the plans.

In 2013, Ahmedabad (India) developed the first heat action plan in South-East Asia following a deadly heat wave in 2010. The Ahmedabad Heat Action Plan 2018 [66], an update of the 2013 version, describes the heat early warning and heat action for this city based on four key strategies: (1) building public awareness and community outreach, (2) initiating an early warning system and inter-agency coordination, (3) capacity building among health care professionals and (4) reducing heat exposure and promoting adaptive measures (Ahmedabad Municipal Corporation 2018). The alert levels consist of three warning levels when certain temperature thresholds are surpassed (41–43 °C: yellow alert, 43.1–44.9 °C: orange alert and above 45 °C: red alert). Regarding the action plans, beyond the already mentioned intervention strategies in Europe, the revised plan promotes the use of white and reflective paint on roofs, access to cool drinking water, digital media campaigns, medical professional training, limit heavy work in direct sun or indoors if poorly ventilated, among others.

## 5. Conclusions

Under a warmer climate, extreme heat events such as summer 2003 could happen every other year by the end of the 21st century in Central Europe [2], and even earlier in Southern Europe. Awareness, preparedness and health-prevention actions need to be taken before such extraordinary situation occurs again. Heat-health action plans coordinate the meteorological heat warnings with a system of graded alerts and a plan of intervention and communication strategies. The joint collaboration of institutional agencies and multidisciplinary approaches are essential for a successful development of heat-health warning systems and action plans.

The present work constitutes a state-of-the-art review of 16 European heat-health warning systems and heat-health action plans, based on the existing literature, web search (over the National Meteorological Services websites) and interviews of experts using questionnaires. The aim of this study is to pave the way for future heat-health warning systems, such as the one currently under development in the framework of the Horizon 2020 HEAT-SHIELD project (www.heat-shield.eu), which is geared towards European employers and employees (see Ref. [29] in the present Special Issue for details).

The ways in which HHWSs in Europe deal with heat warnings are very diverse. They use different variables and thresholds to trigger the warnings. The majority of the systems are based on daily mean or maximum temperature, only a few systems consider other relevant variables, such as minimum temperature or humidity. The intervention strategies implemented against heat waves are similar in the considered countries. There are; however, not many studies evaluating the effectiveness of such actions [57]. The diversity of European warning systems makes it difficult to warn on European scales, which might be of interest for some stakeholders. The HEAT-SHIELD platform is a good example of European-wide and user-tailored warning system with a specific objective, namely, to protect workers’ health and productivity [29].

Based on the review presented here, some recommendations for the design of future heat-health warning systems and for the improvement of present systems can be deduced:The experience from previous extreme heat situations shows that a significant proportion of excess summer deaths occurs before the health heat wave alert is triggered, which emphasizes the importance of long-term planning actions as well as pre-alert levels by local authorities and the health sector.The variables that trigger the warnings should present a clear link with the impact under consideration (i.e., mortality rate, productivity losses). Therefore, the effect of all relevant meteorological variables for heat stress (including minimum and maximum temperature, humidity, wind speed, solar radiation) should be taken into account [9].The optimal index to use strongly depends on the purpose of the warning. The combination of several indices and/or different user-tailored thresholds needs to be considered to ensure an effective warning system (see Ref. [71] in the present Special Issue).The thresholds of the warning indicator should allow for short-term (throughout the year) and long-term (under a warmer climate) adaptation to heat.The thresholds should be based on a probability (risk) approach, i.e., they are set by considering the probability of exceeding certain mortality thresholds rather than an absolute number of deaths [24].Good coordination between the meteorological agency and health ministry or agency is necessary. Heat health warning systems should speak with “one voice” [24].Educational and communication strategies are very important to raise the awareness of the hazard, so that the population is prepared when a heat-wave occurs.The regular evaluation of the effectiveness of heat health warning system and associated interventions is advisable [24]. The revision might include, in the long-term, the effect of changes in city architecture, such as including building passive systems and more green spaces.Information should not be limited to the local language. This is particularly relevant in countries receiving many (potentially unacclimatized) tourists in the summer season.

The heterogeneity of the review methods leads to some limitations of this work. The cited websites or reports are available online as of June 2019, but might be subject to change in future. Many websites of the National Weather Services present most of the contents on the local language, making an in-depth understanding difficult. Some detailed information was collected by means of questionnaires, but these did not cover all European countries. Still, this work represents an updated review of heat-health warning systems, providing an overview on the methods and strategies currently implemented in the European countries considered. Learning from the current (heterogeneous) situation can foster future plans for joint initiatives or plans regarding heat in neighboring countries or throughout the continent.

## Figures and Tables

**Table 1 ijerph-16-02657-t001:** Characteristics of the weather forecast system in European countries. The identifier (ID) for each country is used as reference also in Table 2, Table 3 and Table 4.

ID	Country	Provider, Institution	Model System	Spatial Resolution	Temporal Resolution	Lead Time
AU	Austria	Zentralanstalt für Meteorologie und Geodynamik (ZAMG)	INCA (1 km), ALARO (5 km), ECMWF ENS (15 km)	Regional	Several times a day	6 h (INCA), 72 h (ALARO)
BE	Belgium	Royal Meteorological Institute of Belgium (RMI)	ALADIN (deterministic) + RMI EPS (probabilistic, based on AROME and ALARO, 2.5 km)	Provinces	Several times a day	12–48 h, depending on the warning level
EN	England	Met Office and Public Health England (PHE)	Deterministic UK (UKV, 1.5 km) + Ensemble UK (MOGREPS-UK, 2.2 km)	Regional and local	Several times a day	120 h (UKV) to 5 days (MOGREPS-UK)
FR	France	MeteoFrance and Santé Publique France	AROME (1.3 km) + ARPEGE (16 km), deterministic	Regional	Twice a day (6 and 18 h)	36 h (AROME), 4 days (ARPEGE)
DE	Germany	German Weather Service (DWD)	MOSMIX (optimizes and interprets the forecasts of ICON-DWD and IFS-ECMWF)	County level	Daily	2 days (8 days for pre-information)
GR	Greece	Hellenic National Meteorological Service (HNMS)	ECMWF deterministic and ENS model	Regional (16 regions)	Several times a day	3 days
HU	Hungary	Hungarian Meteorological Service (HMS) and National Public Health and Medical Officer Service (ANTSZ)	ECMWF and GFS, both with deterministic and probabilistic versions	Minimum unit is county level	Four times a day	4 days to the public, 8 days to the authorities
IT	Italy (HHWS Rome)	Servizio Meteorologico Aeronautica Militare (Italian Meteorological Service), Lazio Health Authority and Department of Epidemiology	COSMO deterministic (5 and 2.2 km, depending on the domain) + COSMO-EPS (probabilistic, 7 and 2.2 km)	Regional capitals and cities with more than 250,000 inhabitants (27 cities)	Several times a day	48–72 h, depending on domain size
NE	Netherlands	National Institute for Public Health and the Environment (RIVM) and Royal Netherlands Meteorological Institute (KNMI)	HIRLAM + HARMONIE (2.5 km) + ECMWF ENS (9 km)	Regional	Several times a day	6 days
NM	North Macedonia	Macedonian Meteorology	-	National and regional	-	2 days
PT	Portugal (HHWS Lisbon)	Portuguese Meteorological Institute, Portuguese National Institute of Health, Portuguese General Health Directorate and Portuguese Civil Protection Service	ALADIN (12 km)	Regional and specific for Lisbon	Daily	3 days
RO	Romania	Public Health Ministry and Meteo Romania	-	-	-	2 days
SL	Slovenia	Slovenian Environment Agency (ARSO)	ECMWF incl. MOS and ensembles + ALADIN	5 regions—approximately 4000 km^2^ each	Daily	5 days
SP	Spain	National Plan joining several institutions (e.g., National Meteorological Agency AEMET, Health Ministry)	HARMONIE-AROME (deterministic) + IFS-ECMWF (probabilistic, longer lead times)	Regions and provinces	Several times a day (9, 11:30 and 23 h)	3 days (HARMONY-AROME), 5 days (IFS-ECMWF)
SW	Sweden	Swedish Meteorological and Hydrological Institute (SMHI)	HARMONIE-AROME (0–48h, deterministic + probabilistic with 9 members) + ECMWF (longer lead times, deterministic and ENS)	40 warning districts of variable size. The warning is limited, if necessary, to a smaller part of the district	Every morning. Rather often updates in the afternoon or in the evening, considering the latest run of the model and observed temperatures.	3–5 days ahead depending on the warning criterion
CH	Switzerland	Federal Office of Meteorology and Climatology (MeteoSwiss)	COSMO CLM deterministic + MOSMIX + ENS ECMWF (longer lead times)	Regional	At least once a day. Updates if necessary.	COSMO 3 days, ENS ECMWF 5 days

INCA: Integrated Nowcasting Through Comprehensive Analysis. ALARO: ALadin–AROme. ALADIN: Aire Limitée Adaptation Dynamique Développement Internationa. AROME: Application of Research to Operations at Mesoscale. IFS: Integrated Forecast System. ECMWF: European Center for Medium-Range Weather Forecasts. ENS: ensemble. GFS: Global Forecast System. Table cells with “-” mean missing information.

**Table 2 ijerph-16-02657-t002:** Characteristics of the warning system.

ID	Heat index	Warning thresholds/alerts levels	Nature of the warning thresholds	Target groups
AU	Perceived Temperature (PT) and Tmin	PT > 35 °C for at least 3 days without night cooling below 20 °C Thresholds are subject to modifications depending on weather in preceding days.	Threshold from epidemiological studies (based on thermophysiological strain) in Germany by DWD adapted to Austria.	Nursing facilities, hospitals and health resorts, childcare facilities (kindergartens, schools, etc.), mobile nursing services, medical chambers and emergency organizations
BE	Tmax, Tmin, Tcumul	Green: No warning, all indices below thresholds. Yellow: Tcumul ≥ 17 °C Orange: 2 day mean with Tmax ≥ 32 °C and Tmin ≥ 20 °C or 3 day mean Tmax ≥ 30 °C and Tmin ≥ 18 °C Red: 3 day mean Tmax ≥ 32 °C and Tmin ≥ 20 °C	-	Belgium Interregional Environment Agency (IR-CEL), authorities
EN	Tmax and Tmin	Region-specific (for Tmax and Tmin). Average thresholds: 30 °C for Tmax and 15 °C for Tmin for at least two consecutive days. Five alert levels (Levels 0–4).	Epidemiological (15–20% increased risk of mortality)	National Health Service, local authorities, social care, other public agencies, professionals working with people at risk, individuals, local communities, voluntary groups.
FR	Combination of Tmin and Tmax averaged over three days (BMI)	3 days mean of Tmax > regionally dependent thresholds and 3 days mean of Tmin > regionally dependent thresholds	Biometeorological	Local authorities, Santé Publique France
DE	PT and Tmin	Thresholds are region-specific and consider acclimatization (previous 30 days). Benchmark: Level 1: 20 °C < PT < 26 °C Level 2: 26 °C < PT < 32 °C Level 3—Strong heat stress: 32 °C < PT < 38 °C & Tmin ≥ 17 °C Level 4—Extreme heat stress: PT ≥ 38 °C & Tmin ≥ 17 °C Only Levels 3 and 4 are relevant for heat warnings.	Epidemiological research, based on thermophysiological strain	General public, health system, elderly, people living socially isolated, people needing care, obese persons, chronically ill, working people outdoors, homeless, babies and infants.
GR	Tmax and heat index	a. For Tmax over the northern Greek regions: 35 °C ≤ Tmax < 39 °C (yellow), 39 °C ≤ Tmax < 42 °C (amber), Tmax ≥ 42 °C (red) b. For Tmax over the Central and Southern parts of Greece: 37 °C ≤ Tmax < 41 °C (yellow), 41 °C ≤ Tmax < 44 °C (amber), Tmax ≥ 44 °C (red) c. For Tmax over the islands: 33 °C ≤ Tmax < 37 °C (yellow), 37 °C ≤ Tmax ≤ 40 °C (amber), Tmax ≥ 40 °C (red) Heat index used as a supplementary tool in order to enhance the conclusions from the consideration of each synoptic situation. No thresholds related.	Climatological percentiles and the respective literature	General Public. Other target groups are managed by the Ministry of Health
HU	Tmean	Yellow: 25 °C > Tmean > 27 °C Orange: 27 °C > Tmean > 29 °C Red: Tmean ≥ 29 °C	Epidemiological, link to mortality	National Public Health and Medical Officer Service
IT	Tappmax and “air mass-based approach” in larger cities	City-specific	Epidemiological, related to excess of mortality	Ministry of Health, local health authorities, local civil protection, stakeholders (hospitals, retirement homes etc.) GPs, health resorts, media, registered individuals
NE	Tmax	more than 10% probability of 4 or more days with Tmax > 27 °C	Practical: not too many warnings in a year	Elderly, people in care institutions, and chronically ill and overweight people. Public health services, trade associations and the Dutch Red Cross
NM	Tmax	Monthly thresholds for each of the 4 phases for 13 cities in 6 regions from May to September	-	Retirement homes, GPs, workers.
PT	For heat wave definition: TemperatureFor alert levels: ÍCARO index	Heat wave: Temperature > 32 °C for at least 2 days Level 1 & 2: ÍCARO index < 0.31 Level 3: ÍCARO index 0.31–0.93 Level 4 (Heat-wave alert): ÍCARO index > 0.93	Epidemiological	Portuguese National Institute of Health, the Portuguese Meteorological Institute, Portuguese General Health Directorate and the Portuguese Civil Protection Service
RO	Tmax	Alert: Tmax 35–38 °C, Maximum response: Tmax 35–40 °C	-	-
SL	Tmean and Tmax	Yellow: Tmax > 31 °C Orange: Tmax > 34 °C and/or Tmean > 26 °C Red: Tmax > 37 °C and/or Tmean > 28 °C Tresholds are regionally-based	Climatology-based percentiles	Public (general population), civil defense in case of orange or red warning
SP	Heat wave: TmaxHHAP: Tmax and Tmin	Heat wave: At least 3 days with at least 10% of stations in the region with Tmax > 95th percentile. Warnings for single-day events (yellow | orange | red corresponding to different Tmax for each region: North 34 | 37 | 40; Center and Med. 36 | 39 | 42; South 38 | 40 | 44). HHAP: Level 1: 1–2 days Tmax and Tmin above threshold simultaneously Level 2: 3–4 days above threshold Level 3: 5 days above threshold	Heat wave: Climatological percentiles for each region. HHAP: temperature triggering mortality.	General population, Civil Protection, Maritime Rescue Centers, Department of Traffic, Military Unit for Emergencies, Red Cross, other Met Services, government and ministries, media.
SW	Tmax	Heat warning class 1: Tmax ≥ 30 °C for 3 or 4 days Heat warning class 2: Tmax ≥ 30 °C for at least 5 days or Tmax > 33 °C for at least 3 days. Notification: Tmax ≥ 26 °C for at least 3 days. Risk: same thresholds as for the class 2 warning, but the risk concept is used when the forecast is more uncertain. “notification” and “risk” are not considered warnings	Epidemiological, based on the relation between temperature and rate of mortality.	General public and health providers as hospitals, elderly care etc.
CH	Heat index	Level 3 (orange): at least 3 days with HI > 90 Level 4 (red): at least 5 days with HI > 93	-	Construction industry, the trade unions (e.g., UNIA) and cantonal authorities

Tmax: daily maximum temperature. Tmin: daily minimum temperature. Tcumul: the sum of the difference between Tmax and 25 °C for the 5 coming days (only the positive values are taken into account). PT: perceived temperature (combination of temperature, radiation, wind, humidity). BMI: Bio Meteorological Indicator. Tappmax: maximum apparent temperature. GPs: general practitioners. ÍCARO index: (number of expected deaths with the effect of heat (Yt)/number of expected deaths without the effect of heat). HI: heat index.

**Table 3 ijerph-16-02657-t003:** Characteristics of the heat-health action plans.

ID	Component ofa HHAP (Yes/No)	If Component of a HAP, Intervention Strategies	Evaluation and Revision
AU	Yes	Level 0—Long-term development and planning for the summer: Elaboration and updating of information material and ensure information flow with stakeholders and other administrations. Level 1—protection during summer, between heat waves: preparation of information (general public and specific target groups), check and update emergency phone list. Level 2—during heat wave: - Transmission of heat warnings to umbrella organizations and sponsors of care facilities for the elderly andparticularly heat-sensitive population groups. - Management of hospitals, residential and care facilities and mobile services: - Increased attention to signs of heat-related illnesses. - Recording and documenting room-related heat stress (room temperature, solar radiation) and taking acute adaptation measures (ventilation behaviour, sun protection, changing the occupied zone) - Ensure sufficient availability and use of suitable beverages, monitor liquid balance of endangered persons. - Support cooling of the body (e.g., showers, hand and foot baths) and appropriate clothing, bed linen. - Ensure appropriate nutrition and food safety. - Avoidance of outdoor activities at peak times. - Family and neighborly contacts consciously activate - Phone line available for the population.	Irregularly
BE	Yes	Public cooling areas, phone hotline. -	
EN	Yes	From long-term planning for severe heat, through summer and heatwave preparedness, to a major national emergency. Social and healthcare services target specific actions at high-risk groups. Potential discontinuation of public or sporting events, closure of schools, provision of local cool centers, reduce urban heat & deteriorating air quality by minimizing unnecessary transport and energy use. Implications for trains: staged preventative measures at 22 °C, extreme precautions at 36 °C, measures to prevent track buckling.Bottled water supply.	Annually
FR	Yes	Upscale hotline staff. Track and support homeless (t-shirt, water, sun-cap, map of drinking fountains, emergency shelter venues). Voluntary registry of vulnerable people and monitoring (NGO, including Red Cross). Installation and maintenance of air-conditioned in residential care. Installation of public water provision or open up public swimming pools.	Annually
DE	Yes, in some federal states of Germany	Depending on the federal states of Germany. Overall phone hotline and information campaign. Example in Berlin: “heat bus”, supplying refreshments, sun protection and information.	Annually
GR	General Secretariat for Civil Protection responsible for actions	-	No
HU	Yes	Forecasted heat wave: info to health care system and general public. During heatwave: provide portable water in public places, water roads and parks, monitor water supply and quality, planned disruption of electricity, special rules for employers and restrictions for public transport. Extraordinary measures: increase hospital beds, ambulance units, hospital staff, cool bodies at morgues, extend opening hours public air-conditioned places and pools, defer non-essential surgery.	-
IT	Yes	Active monitoring of vulnerable groups by GPs, social workers, volunteers (phone calls & home visits by GPs). Activation of emergency protocols in care and retirement homes and in hospitals (postpone non-emergency surgery, discharge planning, staff rotation restrictions, increased hospital beds). Educational campaigns.	-
NE	Yes	Several organizations warn their target groups and regional contacts. Measures to limit the impact of hot weather conditions provided through the heat plan.	Regular meetings to discuss the method of the heat plan and the effects
NM	Yes	Monitoring, information provided to retirement homes and GPs, installation or maintenance of public drinking fountains & springs, education to public. Phase 1: preventative measures media campaign, home visits to elderly, socially isolated and homeless (red cross); phone line. Phase 2: supply food to elderly and at risk media alert, specific measure for health care preparedness, protection measures for occupationally heat exposed workers; including activating redistribution in residential settings to air-conditioned rooms, extra staff on hotline. Phase 3: Emergency, led by National Crises Management Center	Annually
PT	Yes	Public health emergency telephone used as hotline and reinforced with nursing personnel. Increase capacity of health care services, upscale staff for telephone hotline. Info to public, authorities, health sectors & media. Activate local refuge shelters, monitor need for transportation to places of refuge; notify most vulnerable; increase capacity of health care services.	In specific, extreme years.
RO	Yes	Monitoring and general advice to public and health institutions. Phone hotline; daily information to health ministry, health authorities; monitors sanitary, water, food prep, med storage, outreach people with social dependence. Emergency response: increase support to ambulance or emergency services.	-
SL	No	-	-
SP	Yes	Levels 0, 1: information (media, social services, institutions) and monitoring. Level 2: intensify communication (hospitals and social services), evaluation of specific measures. Level 3: recommendations to population under risk, evaluation by the Centre of Warnings and Health Emergencies.	At least once a year
SW	The SMHI is only in charge of the heat warnings, not the actions.	It is up to each county to develop strategies.	All class 2 warnings are evaluated. The whole warning system will be revised in the next few years with the introduction of impact-based warnings and flexible warning districts.
CH	Yes, cantonal heat plans	The strategies depend on the cantonal authorities. For instance, in Ticino: -During the period of health surveillance, from 1 June to 15 September, constant monitoring of the weather. -In the event of a heat wave: continuous communication with the main partners (Service for the Protection of the Population, Labour Inspectorate Office, municipalities, homes for the elderly, home care and assistance services, representatives of the world of work) and health monitoring in collaboration with the emergency department of the Ente Ospedaliero Cantonale	Feedback from cantons after each warning and annual warnings’ conference.

GPs: general practitioners.

**Table 4 ijerph-16-02657-t004:** Communication strategies of the heat warnings.

ID	Language	Notification System to Target Groups	Notification System to Stakeholders	Source	Website
AU	German	Website	Internally, all warnings above a certain category are subject to a certified process, which determines which bodies have to be informed and in which form. Email-Newsletter with detailed forecasts, tips, links etc. to nursing homes, kindergardens, hospitals, civil protection, firefighters, red cross.	[67,68]	https://warnungen.zamg.at
BE	Belgian	Website	-	www.meteo.be/en/weather/warnings/legend-heat	www.meteo.be/en/weather/warnings/overview-belgium
EN	English	Media, website, emails, social media, leaflet (via pharmacies, GPs, national health system, advice centres, hospitals, care homes)	Email-Newsletter	[53] www.metoffice.gov.uk/public/weather/heat-health	www.metoffice.gov.uk/weather/warnings-and-advice
FR	French	Communicated to the media and the general population by MétéoFrance through a vigilance map	-	[69,70]	http://vigilance.meteofrance.com/
DE	German	Email-Newsletter, website, Apps, radio, internet, newspaper, television	Email-Newsletter	[41] www.dwd.de/EN/weather/warnings/warnings_node.html http://media.repro-mayr.de/52/113152.pdf#page=40	www.dwd.de/DE/wetter/warnungen_gemeinden/warnWetter_node.html
GR	Greek	Website, twitter	The Ministry of Health issues the notifications to all the stakeholders.	-	www.emy.gr/emy/en
HU	Hungarian and English, via web and Meteoalarm	Website, mobile app (Meteora)	E-mails to the National Public Health and Medical Officer Service	[27] www.antsz.hu	www.met.hu/en/idojaras/veszelyjelzes/
IT	Italian	City-specific warning bulletins are distributed at both national and local level and published online.Information through national/local help-lines and via the media. Diffusion of warnings via the media, Ministry of Health/Civil Protection websites.	Warning bulletins published daily on the website and sent to stakeholders. Flyers in centres for elderly and public places, local pharmacies health centres and GPs.	[24]; www.ccm-network.it/documenti_Ccm/prg_area3/Piano_caldo_2009-2012/report/4semestre/Allegato4.2_poster-Eph2011.pdf http://medcof.aemet.es/images/doc_events/medcof7/docMedcof7/presentaciones/MedCOF7_DeDonato_SSRL.pdf www.euro.who.int/__data/assets/pdf_file/0008/96965/E82629.pdf	www.meteoam.it
NE	Dutch	Letters, media (press release), emails etc.	Public health services, trade associations and the Dutch Red Cross warn their supporters and regional contacts.Practical information on measures in the toolkit ‘Hitte’ (‘heat’). It contains: list of questions and answers, public brochure, sample letters, press announcement, text of the National Heat Plan. It is aimed at environmental health experts, environmental nurses, medical environmentalists, communication staff, etc.	www.rivm.nl/publicaties/nationaal-hitteplan-versie-2015 www.knmi.nl/kennis-en-datacentrum/uitleg/weermodellen www.rivm.nl/hitte	www.knmi.nl/nederland-nu/weer/waarschuwingen
NM	Macedonian	-	-	[27]	https://uhmr.gov.mk
PT	Portuguese	-	-	[24,49] www.euro.who.int/__data/assets/pdf_file/0008/96965/E82629.pdf	www.ipma.pt/en/otempo/prev-sam
RO	Romanian	-	Daily information to health ministry and health authorities.	[27]	www.meteoromania.ro/avertizari
SL	Slovene	ARSO web pages, Twitter, Facebook, radio, TV	Website, Twitter, Facebook, radio, TV. Civil defense by E-mail.		www.meteo.si/met/sl/warning
SP	Spanish (and CAP in English)	Website (maps, bulletins and CAP), social media (especially Twitter), Open Data server for further analyses.	Direct/automatic distribution to public bodies. CAP and bulletins.	www.aemet.es/documentos/es/eltiempo/prediccion/avisos/plan_meteoalerta/plan_meteoalerta.pdf www.aemet.es/documentos/es/eltiempo/prediccion/avisos/plan_meteoalerta/umbrales_niveles_aviso.pdf	www.aemet.es/es/eltiempo/prediccion/avisos
SW	Swedish and English	Authorities and institutions in the health sector and in the elderly care can register to receive emails when heat warnings are issued.	Stakeholders are notified by email, via mobile application, website, social media and/or via early warning telephone conference.	-	www.smhi.se/en/weather/sweden-weather/warnings
CH	German/French/Italian and the SMS also inEnglish.	MeteoSwiss app and website.Postal code subscription via app for receiving region-specific warnings.	The dissemination of the warning message to the stakeholders (hospitals, retirement homes, etc...) is handled differently depending on the region. In Ticino and French-speaking Switzerland, some heat products are sent daily in summer (forecasts of Tmax, Tmin).	www.meteoswiss.admin.ch/home/weather/gefahren.html https://www.nccs.admin.ch/nccs/de/home/regionen/kantone.html	www.naturgefahren.ch

GPs: general practitioners. CAP: common alerting protocol.

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
