# Peer review of "Overview of Existing Heat-Health Warning Systems in Europe"

_ijerph, 2019, doi:10.3390/ijerph16152657_

Round 1

Reviewer 1 Report

The paper performs a review of 16 European heat-health warning systems (HHWS). It highlights the different steps of each HHWS and the main differences between countries. It is a very useful paper for public health stakeholders and researcher to implement and improve HHWS across the world.

I have very few minor comments on the paper and thus think it only necessitates minor revision.

Major comments

1.       I think the paper misses a more extensive overview of the statistical methods used to determine thresholds. Information given in section 4.2 (especially p.9, l.22 and in Table 2) is a bit too vague on this subject in my opinion. The statistical method considered to determine thresholds is an important part of HHWS are a discussion on the subject would help researchers and public health stakeholder identify and address current shortcomings.

2.       P.16, L.29-36: Isn’t the HHWS of United States based on airmass categories, in the same fashion as Italy?

3.       P.20, L.40: I am not sure I understand well what the authors mean by “the threshold should be based on a probability (risk) approach”. Does it refer to statistical models or to the necessity to consider what statisticians would call soft thresholds, i.e. levels of confidence to face adverse condition, by opposition to a hard threshold?

4.       Are there local specificities that could have impacted the HHWS and HHAP implemented in the different countries? For instance, does the prominence of air conditioning in some countries affect the threshold or actions performed?

Minor comments

-          P.1, L.32: Do authors mean “increase” instead of “increment”?

-          In Table 1, for France: I think the INVS recently changed its name into Santé Publique France (Public Health France).

Reviewer 2 Report

The manuscript entitled “Overview of existing heat-health warning systems in Europe” deals with a comparison between the existing European warning systems, their characteristics, methods of warnings, and their issued recommendations. Then, the authors state a list of recommendations based on the reviewed European warnings.

General comments:

Before the warning systems are described, the authors should set the main features to consider a heat warning in Europe. Some descriptions in the Introduction section is carried out, but are not enough. What are the most common values of temperature, relative humidity, solar irradiance and other factors to consider the presence of a heat wave? There is a large literature review regarding this subject.

In the recommendations, in my opinion, there are two more topics: in the long-term, the use of building passive systems useful during the cooling season; and the countering of the urban heat island effect, through urban planning such as green areas, urban vegetation etc. I would want the authors to discuss this in a deeper manner.

Specific comments:

Page 2, line 28: place a colon jus after “HHWS requires”.

Page 3, line 6: why only publications in English, Spanish and German were considered?

Page 16, line 34: please make the conversion from °F to °C.

Author Response

Please, see attached file.

Reviewer 3 Report

This paper is a very good one about the actual European start-of-art about HHAPs (that is a very good thing to have a deep panorama of the European context into a holistic way of view). That is a very interesting and didactic material which surely needs to entangle HHAPs and HHWSs.

Only one observation (a suggestion) at page 2, line 9: " Even under a 2°C warmer world, the number of heat-related deaths in urban areas could increase by a factor of 2 to 3 [19]". - Please, show to the reader what means a factor 2 and a factor 3.

This is a very interesting paper which seeks for better actions and better systems against heat waves, global warming and human resilience levels. 

Author Response

Please, see attached file.
